# Convolutional Tensor-Train LSTM for Long-Term Video Prediction

## Abstract

Long-term video prediction is highly challenging since it entails simultaneously capturing spatial and temporal information across a long range of image frames. Standard recurrent models are ineffective since they are prone to error propagation and cannot effectively capture higher-order correlations. A potential solution is to extend to higher-order recurrent models. However, such a model requires a large number of parameters and operations, making it intractable to learn and prone to overfitting in practice. In this work, we propose *Convolutional Tensor-Train LSTM* (Conv-TT-LSTM), which learns higher-order *Convolutional Long Short-Term Memory* (ConvLSTM) efficiently using *Convolutional Tensor-Train Decomposition* (CTTD). Our proposed model naturally incorporates higher-order spatio-temporal information with low memory and computational requirements by efficient low-rank tensor-train representations. We evaluate our model on Moving-MNIST and KTH datasets and show improvements over standard ConvLSTM and other ConvLSTM-based approaches, but with much fewer parameters.

## 1 Introduction

Understanding dynamics of videos and performing long-term predictions of the future is a highly challenging problem. It entails learning complex representation of real-world environment without external supervision. This arises in a wide range of applications, including autonomous driving, robot control (Finn & Levine, 2017), or other visual perception tasks like action recognition or object tracking (Alahi et al., 2016). However, long-term video prediction remains an open problem due to high complexity of the video contents. Therefore, prior works mostly focus on next or first few frames prediction (Lotter et al., 2016; Finn et al., 2016; Byeon et al., 2018).

Many recent video models use *Convolutional LSTM* (ConvLSTM) as a basic block (Xingjian et al., 2015), where spatio-temporal information is encoded as a tensor explicitly in each cell. In ConvLSTM networks, each cell is a first-order recurrent model, where the hidden state is updated based on its immediate previous step. Therefore, they cannot easily capture higher-order temporal correlations needed for long-term prediction. Moreover, they are highly prone to error propagation.

Various approaches have been proposed to augment ConvLSTM, either by modifying networks to explicitly modeling motion (Finn et al., 2016), or by integrating spatio-temporal interaction in ConvLSTM cells (Wang et al., 2017; 2018a). These approaches are often incapable of capturing long-term dependencies and produce blurry prediction.

Another direction to augment ConvLSTM is to incorporate a higher-order RNNs (Soltani & Jiang, 2016) inside each LSTM cell, where its hidden state is updated using multiple past steps. However, a higher-order model for high-dimensional data (e.g. video) requires a huge number of model parameters, and the computation grows exponentially with the order of the RNNs. A principled approach to address the curse of dimensionality is tensor decomposition, where a higher-order tensor is compressed into smaller core tensors (Anandkumar et al., 2014). Tensor representations are powerful since they retain rich expressivity even with a small number of parameters. In this work, we propose a novel convolutional tensor decomposition, which allows for compact higher-order ConvLSTM.

**Contributions.** We propose *Convolutional Tensor-Train LSTM* (Conv-TT-LSTM), a modification of ConvLSTM, to build a higher-order spatio-temporal model. **(1)** We introduce *Convolutional Tensor-Train Decomposition* (CTTD) that factorizes a large convolutional kernel into a chain of

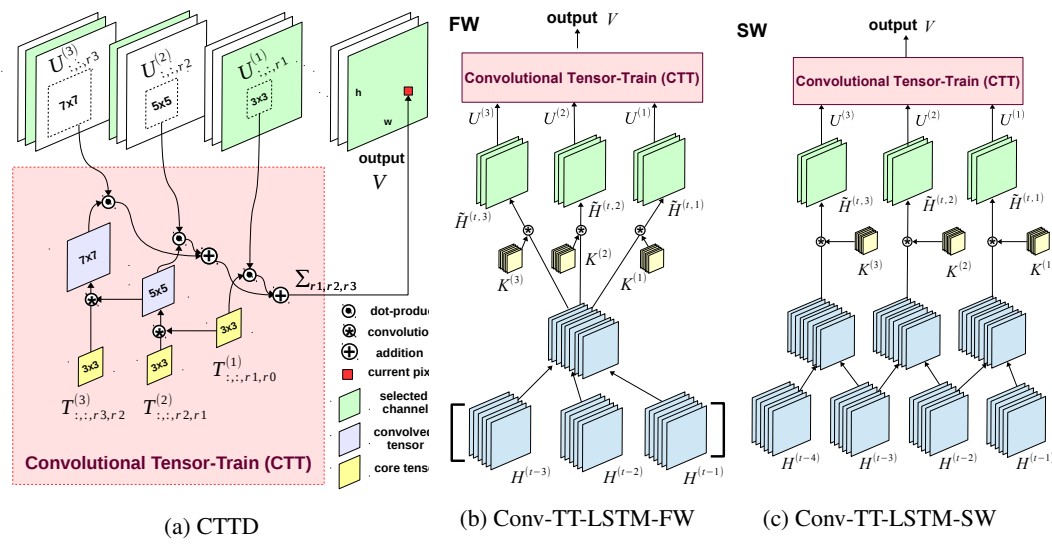

Figure 1: Illustration of **(a) convolutional tensor-train** (Eqs. (5) and (6)) and the difference between **convolutional tensor-train LSTM (b)** Fixed window version (Eqs. (11a) and (10)) and **(c)** Sliding window version (Eqs. (11b) and (10)). The *fixed window version* use all steps to compute each input to convolutional tensor-train, while *sliding window version* uses a window of steps for each input.

smaller tensors. **(2)** We integrate CTTD into ConvLSTM and propose Conv-TT-LSTM, which learns long-term dynamics in video sequence with a small number of model parameters. **(3)** We propose two versions of Conv-TT-LSTM: Fixed Window (FW) and Sliding Window (SW) (See Figures 1b and 1c), and we found that the SW version performs better than the FW one. **(4)** We found that training higher-order tensor models is not straightforward due to gradient instability. We present several approaches to overcome this such as good learning schedules and gradient clipping. **(5)** In the experiments, we show our proposed Conv-TT-LSTM consistently produces sharp prediction over a long period of time for both Moving-MNIST-2 and KTH action datasets. Conv-TT-LSTM outperforms the state-of-the-art PredRNN++ (Wang et al., 2018a) in LPIPS (Zhang et al., 2018) by $\mathbf{0.050}$ on the Moving-MNIST-2 and $\mathbf{0.071}$ on the KTH action dataset, with $\mathbf{5.6}$ times fewer parameters. Thus, we obtain best of both worlds: better long-term prediction and model compression.

## 2  RELATED WORK

**Tensor Decomposition**  In machine learning, tensor decompositions, including *CP decomposition* (Anandkumar et al., 2014), *Tucker decomposition* (Kolda & Bader, 2009), and *tensor-train decomposition* (Oseledets, 2011), are widely used for dimensionality reduction (Cichocki et al., 2016) and learning probabilistic models (Anandkumar et al., 2014). In deep learning, prior works focused on their application in model compression, where the parameters tensors are factorized into smaller tensors. This technique has been used in compressing convolutional networks (Lebedev et al., 2014; Kim et al., 2015; Novikov et al., 2015; Su et al., 2018; Kossaifi et al., 2017; Kolbeinsson et al., 2019; Kossaifi et al., 2019), recurrent networks (Tjandra et al., 2017; Yang et al., 2017) and transformers (Ma et al., 2019). Specifically, Yang et al. (2017) demonstrates that the accuracy of video classification increases if the parameters in recurrent networks are compressed by tensor-train decomposition (Oseledets, 2011). Yu et al. (2017) used tensor-train decomposition to constrain the complexity of higher-order LSTM, where each next step is computed based on the outer product of previous steps. While this work only considers vector input at each step, we extend their approach to higher-order ConvLSTM, where each step also encodes spatial information.

**Video Prediction**  Prior works on video prediction have focused on several directions: predicting short-term video (Lotter et al., 2016; Byeon et al., 2018), decomposing motion and contents (Finn et al., 2016; Villegas et al., 2017; Denton et al., 2017; Hsieh et al., 2018), improving the objective function Mathieu et al. (2015), and handling diversity of the future (Denton & Fergus, 2018;

Babaeizadeh et al., 2017; Lee et al., 2018). Many of these works use Convolutional LSTM (ConvL-STM) (Xingjian et al., 2015) as a base module, which deploys 2D convolutional operations in LSTM to efficiently exploit spatio-temporal information. Finn et al. (2016) used ConvLSTM to model pixel motion. Some works modified the standard ConvLSTM to better capture spatio-temporal correlations (Wang et al., 2017; 2018a). Wang et al. (2018b) integrated 3D convolutions into ConvLSTM. In addition, current cell states are combined with its historical records using self-attention to efficiently recall the history information. Byeon et al. (2018) applied ConvLSTM in all possible directions to capture full contexts in video and also demonstrated strong performance using a deep ConvLSTM network as a baseline. This baseline is adapted to obtain the base architecture in the present paper.

## 3 TENSOR-TRAIN DECOMPOSITION AND SEQUENCE MODELING

The goal of tensor decomposition is to represent a higher-order tensor as a set of smaller and lower-order *core tensors*, with fewer parameters while preserve essential information. In Yu et al. (2017), *tensor-train decomposition* (Oseledets, 2011) is used to reduce both parameters and computations in higher-order recurrent models, which we review in the first part of this section.

However, the approach in Yu et al. (2017) only considers recurrent models with vector inputs and cannot cope with image inputs directly. In the second part, we extend the standard *tensor-train decomposition* to *convolutional tensor-train decomposition* (CTTD). With CTTD, a large convolutional kernel is factorized into a chain of smaller kernels. We show that such decomposition can reduce both parameters and operations of higher-order spatio-temporal recurrent models.

**Standard Tensor-train decomposition**  Given an $m$-*order* tensor $\mathcal{T} \in \mathbb{R}^{I_1 \times \cdots \times I_m}$, where $I_l$ is the *dimension* of its $l$-th order, a standard tensor-train decomposition (TTD) (Oseledets, 2011) factorizes the tensor $\mathcal{T}$ into a set of $m$ *core tensors* $\{\mathcal{T}^{(l)}\}_{l=1}^{m}$ with $\mathcal{T}^{(l)} \in \mathbb{R}^{I_l \times R_l \times R_{l+1}}$ such that

$$\mathcal{T}_{i_1,\cdots,i_m} \triangleq \sum_{r_1=1}^{R_1} \cdots \sum_{r_{m-1}=1}^{R_{m-1}} \mathcal{T}_{i_1,1,r_1}^{(1)} \, \mathcal{T}_{i_2,r_1,r_2}^{(2)} \cdots \mathcal{T}_{i_m,r_{m-1},1}^{(m)} \tag{1}$$

where *tensor-train ranks* $\{R_l\}_{l=0}^{m}$ (with $R_0 = R_m = 1$) control the number of parameters in the *tensor-train format* Eq.(1). With TTD, the original tensor $\mathcal{T}$ of size $(\prod_{l=1}^{m} I_l)$ is compressed to $(\sum_{l=1}^{m} I_l R_{l-1} R_l)$ entries, which grows linearly with the order $m$ (assuming $R_l$'s are constant). Therefore, TTD is commonly used to approximate higher-order tensors with fewer parameters.

The sequential structure in tensor-train decomposition makes it particularly suitable for sequence modeling (Yu et al., 2017). Consider a higher-order recurrent model that predicts a scalar output $v \in \mathbb{R}$ based on the outer product of a sequence of input vectors $\{u^{(l)} \in \mathbb{R}^{I_l}\}_{l=1}^{m}$ according to:

$$v = \left\langle \mathcal{T}, \left( u^{(1)} \otimes \cdots \otimes u^{(m)} \right) \right\rangle = \sum_{i_1=1}^{I_1} \cdots \sum_{i_m=1}^{I_m} \mathcal{T}_{i_1,\cdots,i_m} \, u_{i_1}^{(1)} \cdots u_{i_m}^{(m)} \tag{2}$$

This model is intractable in practice since the number of parameters in $\mathcal{T} \in \mathbb{R}^{I_1 \times \cdots I_m}$ (and therefore computational complexity of Eq. (2)) grows exponentially with the order $m$. Now suppose $\mathcal{T}$ takes a tensor-train format as in Eq. (1), we prove in Appendix A that (2) can be efficiently computed as

$$v_{r_l}^{(l)} = \sum_{i_l=1}^{I_l} \sum_{r_{l-1}=1}^{R_l} \mathcal{T}_{i_l,r_{l-1},r_l}^{(l)} \, v_{r_{l-1}}^{(l-1)} \, u_{i_l}^{(l)}, \; \forall l \in [m] \tag{3}$$

where the vectors $\{v^{(l)} \in \mathbb{R}^{R_l}\}_{l=1}^{m}$ are the intermediate steps, with $v^{(0)} \in \mathbb{R}$ initialized as $v^{(0)} = 1$, and final output $v = v^{(m)}$. Notice that the higher-order tensor $\mathcal{T}$ is never reconstructed in the sequential process in Eq. (3), therefore both space and computational complexities grow linearly (not exponentially compared to Eq. (2))with the order $m$ assuming all tensor-train ranks are constants.

**Convolutional Tensor-Train Decomposition**  A convolutional layer in neural network is typically parameterized by a 4-th order tensor $\mathcal{T} \in \mathbb{R}^{K \times K \times R_m \times R_0}$, where $K$ is the kernel size, $R_m$ and $R_0$ are the number of input and output channels respectively. Suppose the kernel size $K$ takes the form $K = m(k-1) + 1$ (e.g. $K = 7$ and $m = 3$, $k = 3$), a convolutional tensor-train decomposition

(CTTD) factorizes $\mathcal{T}$ into a set of $m$ core tensors $\{\mathcal{T}^{(l)}\}_{l=1}^m$ with $\mathcal{T}^{(l)} \in \mathbb{R}^{k \times k \times R_l \times R_{l-1}}$ such that

$$\mathcal{T}_{:,:,r_m,r_0} \triangleq \sum_{r_1=1}^{R_1} \cdots \sum_{r_{m-1}=1}^{R_{m-1}} \mathcal{T}_{:,:,r_1,r_0}^{(1)} * \mathcal{T}_{:,:,r_2,r_1}^{(2)} * \cdots * \mathcal{T}_{:,:,r_m,r_{m-1}}^{(m)} \tag{4}$$

where $*$ denotes convolution between 2D-filters, and $\{R_l\}_{l=1}^m$ are the *convolutional tensor-train ranks* that control the complexity of the *convolutional tensor-train format* in Eq. (4). With CTTD, the number of parameters in the decomposed format reduces from $K^2 R_0 R_m$ to $\left(\sum_{l=1}^m k^2 R_{l-1} R_l\right)$.

Similar to standard TTD, its convolutional counterpart can also be used to compress higher-order spatio-temporal recurrent models with convolutional operations. Consider a model that predicts a 3-rd order feature $\mathcal{V} \in \mathbb{R}^{H \times W \times R_0}$ based on a sequence of 3-rd features $\{\mathcal{U}^{(l)} \in \mathbb{R}^{H \times W \times R_l}\}_{l=1}^m$ (where $H, W$ are height/width of the features and $R_l$ is the number of channels in $\mathcal{U}^{(l)}$) such that

$$\mathcal{V}_{:,:,r_0} = \sum_{l=1}^m \mathcal{W}_{:,:,r_l,r_0}^{(l)} * \mathcal{U}_{:,:,r_l}^{(l)}, \text{ with } \mathcal{W}^{(l)} = \mathsf{CTTD}\left(\{\mathcal{T}^{(l)}\}_{l=k}^m\right), \forall l \in [m] \tag{5}$$

where $\mathcal{W}^{(l)} \in \mathbb{R}^{[l(k-1)+1] \times [l(k-1)+1] \times R_l \times R_0}$ is the corresponding weights tensor for $\mathcal{U}^{(l)}$. Suppose each $\mathcal{W}^{(l)}$ takes a convolutional tensor-train format in Eq. (4), we prove in Appendix A that the model in Eq. (5) can be computed sequentially similarly without reconstructing the original $\mathcal{W}^{(l)}$'s:

$$\mathcal{V}_{:,:,r_{l-1}}^{(l-1)} = \sum_{r_l=1}^{R_l} \mathcal{T}_{:,:,r_l,r_{l-1}}^{(l)} * \left(\mathcal{V}_{:,:,r_l}^{(l)} + \mathcal{U}_{:,:,r_l}^{(l)}\right), \ \forall l \in [m] \tag{6}$$

where $\{\mathcal{V}^{(l)} \in \mathbb{R}^{H \times W \times R_l}\}_{l=1}^m$ are intermediate results of the sequential process, where $\mathcal{V}^{(m)} \in \mathbb{R}^{H \times W \times R_m}$ is initialized as all zeros and final prediction $\mathcal{V} = \mathcal{V}^{(0)}$. The operations in Eq. (5) is illustrated in Figure 1a. In this paper, we denote the Eq.(5) simply as $\mathcal{V} = \mathsf{CTT}(\{\mathcal{T}^{(l)}\}_{l=1}^m, \{\mathcal{U}^{(l)}\}_{l=1}^m)$.

# 4 CONVOLUTIONAL TENSOR-TRAIN LSTM NETWORKS

Convolutional LSTM is a basic block for most recent video forecasting models (Xingjian et al., 2015), where the spatial information is encoded explicitly as tensors in the LSTM cells. In a ConvLSTM network, each cell is a first-order Markov model, i.e. the hidden state is updated based on its previous step. In this section, we propose convolutional tensor-train LSTM, where convolutional tensor-train is incorporated to model multi-steps spatio-temporal correlation explicitly.

*Notations.* In this section, the symbol $*$ is overloaded to denote convolution between higher-order tensors. For instance, given a 4-th order weights tensor $\mathcal{W} \in \mathbb{R}^{K \times K \times S \times C}$ and a 3-rd order input tensor $\mathcal{X} \in \mathbb{R}^{H \times W \times S}$, $\mathcal{Y} = \mathcal{W} * \mathcal{X}$ computes a 3-rd output tensor $\mathcal{Y} \in \mathbb{R}^{H \times W \times T}$ as $\mathcal{Y}_{:,:,c} = \sum_{s=1} \mathcal{W}_{:,:,s,c} * \mathcal{X}_{:,:,s}$. The symbol $\circ$ is used to denote element-wise product between two tensors, and $\sigma$ represents a function that performs element-wise (nonlinear) transformation on a tensor.

**Convolutional LSTM** Xingjian et al. (2015) extended *fully-connected LSTM* (FC-LSTM) to Convolutional LSTM (ConvLSTM) to model spatio-temporal structures within each recurrent unit, where all features are encoded as 3-rd order tensors with dimensions (height $\times$ width $\times$ channels) and matrix multiplications are replaced by convolutions between tensors. In a ConvLSTM cell, the parameters are characterized by two 4-th order tensors $\mathcal{W} \in \mathbb{R}^{K \times K \times S \times 4C}$ and $\mathcal{T} \in \mathbb{R}^{K \times K \times C \times 4C}$, where $K$ is the kernel size of all convolutions and $S$ and $C$ are the numbers of channels of the input $\mathcal{X}^{(t)} \in \mathbb{R}^{H \times W \times S}$ and hidden states $\mathcal{H}^{(t)} \in \mathbb{R}^{H \times W \times C}$ respectively. At each time step $t$, a ConvLSTM cell updates its hidden states $\mathcal{H}^{(t)} \in \mathbb{R}^{H \times W \times C}$ based on the previous step $\mathcal{H}^{(t-1)}$ and the current input $\mathcal{X}^{(t)}$, where $H$ and $W$ are the height/width that are the same for $\mathcal{X}^{(t)}$ and $\mathcal{H}^{(t)}$.

$$\left[\mathcal{I}^{(t)}; \mathcal{F}^{(t)}; \tilde{\mathcal{C}}^{(t)}; \mathcal{O}^{(t)}\right] = \sigma\left(\mathcal{W} * \mathcal{X}^{(t)} + \mathcal{T} * \mathcal{H}^{(t-1)}\right) \tag{7}$$

$$\mathcal{C}^{(t)} = \tilde{\mathcal{C}}^{(t)} \circ \mathcal{I}^{(t)}; \quad \mathcal{H}^{(t)} = \mathcal{O}^{(t)} \circ \mathcal{C}^{(t)} \tag{8}$$

where $\sigma(\cdot)$ applies sigmoid on the input gate $\mathcal{I}^{(t)}$, forget gate $\mathcal{F}^{(t)}$, output gate $\mathcal{O}^{(t)}$, and hyperbolic tangent on memory cell $\tilde{\mathcal{C}}^{(t)}$. Note that all tensors $\mathcal{C}^{(t)}, \mathcal{I}^{(t)}, \mathcal{F}^{(t)}, \mathcal{O}^{(t)} \in \mathbb{R}^{H \times W \times C}$ are 3-rd order.

**Convolutional Tensor-Train LSTM**   In Conv-TT-LSTM, we introduce a higher-order recurrent unit to capture multi-steps spatio-temporal correlations in LSTM, where the hidden state $\mathcal{H}^{(t)}$ is updated based on its $n$ previous steps $\{\mathcal{H}^{(t-l)}\}_{l=1}^{n}$ with an $m$-order convolutional tensor-train (CTT) as in Eq. (5). Concretely, suppose the parameters in CTT are characterized by $m$ tensors of $4$-th order $\{\mathcal{T}^{(o)}\}_{o=1}^{m}$, Conv-TT-LSTM replaces Eq. (7) in ConvLSTM by two equations:

$$\tilde{\mathcal{H}}^{(t,o)} = f\left(\mathcal{K}^{(o)}, \{\mathcal{H}^{(t-l)}\}_{l=1}^{n}\right), \forall o \in [m] \tag{9}$$

$$\left[\mathcal{I}^{(t)}; \mathcal{F}^{(t)}; \tilde{\mathcal{C}}^{(t)}; \mathcal{O}^{(t)}\right] = \sigma\left(\mathcal{W} * \mathcal{X}^{(t)} + \text{CTT}\left(\{\mathcal{T}^{(o)}\}_{o=1}^{m}, \{\tilde{\mathcal{H}}^{(t,o)}\}_{o=1}^{m}\right)\right) \tag{10}$$

**(1)** Since $\text{CCT}(\{\mathcal{T}^{(l)}\}_{l=1}^{m}, \cdot)$ takes a sequence of $m$ tensors as inputs, the first step in Eq. (9) maps the $n$ inputs $\{\mathcal{H}^{(t-l)}\}_{l=1}^{n}$ to $m$ intermediate tensors $\{\mathcal{H}^{(t,o)}\}_{o=1}^{m}$ with a function $f$. **(2)** These $m$ tensors $\{\tilde{\mathcal{H}}^{(t,o)}\}_{o=1}^{m}$ are then fed into $\text{CCT}(\{\mathcal{T}^{(l)}\}_{l=1}^{m}, \cdot)$ and compute the gates according to Eq. (10).

We propose two realizations of Eq. (9), where the first realization uses a *fixed window* of $\{\mathcal{H}^{(t-l)}\}_{l=1}^{n}$ to compute each $\tilde{\mathcal{H}}^{(t,o)}$, while the second one adopts a *sliding window* strategy. At each step, the Conv-TT-LSTM model computes $\mathcal{H}^{(t)}$ by replacing Eq. (9) by either Eq. (11a) or (11b).

**Conv-TT-LSTM-FW:**     $\tilde{\mathcal{H}}^{(t,o)} = \mathcal{K}^{(o)} * \hat{\mathcal{H}}^{(t,o)} = \mathcal{K}^{(o)} * \left[\mathcal{H}^{(t-n)}; \cdots ; \mathcal{H}^{(t-1)}\right]$ (11a)

**Conv-TT-LSTM-SW:**     $\tilde{\mathcal{H}}^{(t,o)} = \mathcal{K}^{(o)} * \hat{\mathcal{H}}^{(t,o)} = \mathcal{K}^{(o)} * \left[\mathcal{H}^{(t-n+m-l)}; \cdots ; \mathcal{H}^{(t-l)}\right]$ (11b)

In the fixed window version, the previous steps $\{\mathcal{H}^{(l)}\}_{l=1}^{n}$ are concatenated into a $3$-rd order tensor $\hat{\mathcal{H}}^{(t,o)} \in \mathbb{R}^{H \times W \times nC}$, which is then mapped to a tensor $\tilde{\mathcal{H}}^{(t,o)} \in \mathbb{R}^{H \times W \times R}$ by 2D-convolution with a kernel $\mathcal{K}^{(l)} \in \mathbb{R}^{k \times k \times nC \times R}$. And in the sliding window version, $\{\mathcal{H}^{(l)}\}_{l=1}^{n}$ are concatenated into a $4$-th order tensor $\hat{\mathcal{H}}^{(t,o)} \in \mathbb{R}^{H \times W \times D \times C}$ (with $D = n - m + 1$), which is mapped to $\tilde{\mathcal{H}}^{(t,o)} \in \mathbb{R}^{H \times W \times R}$ by 3D-convolution with a kernel $\mathcal{K}^{(l)} \in \mathbb{R}^{k \times k \times D \times R}$. For later reference, we name the model with Eqs.(11a) and (10) as Conv-TT-LSTM-FW and the one with Eqs.(11b) and (10) as Conv-TT-LSTM-SW. Figure 1b and Figure 1c visualize the difference between these two variants.

## 5   EXPERIMENTS

We first evaluate our approach extensively on the synthetic Moving-MNIST-2 dataset (Srivastava et al., 2015). In addition, we use KTH human action dataset (Laptev et al., 2004) to test the performance of our models in more realistic scenario.

**Model Architecture**   All experiments use a stack of 12-layers of ConvLSTM or Conv-TT-LSTM with 32 channels for the first and last 3 layers, and 48 channels for the 6 layers in the middle. A convolutional layer is applied on top of all LSTM layers to compute the predicted frames. Following Byeon et al. (2018), two skip connections performing concatenation over channels are added between (3, 9) and (6, 12) layers. Illustration of the network architecture is included in the appendix. All parameters are initialized by Xavier's normalized initializer (Glorot & Bengio, 2010) and initial states in ConvLSTM or Conv-TT-LSTM are initialized as zeros.

**Evaluation Metrics**   We use two traditional metrics MSE (or PSNR) and SSIM (Wang et al., 2004), and a recently proposed deep-learning based metric LPIPS (Zhang et al., 2018), which measures the similarity between deep features. Since MSE (or PSNR) is based on pixel-wise difference, it favors vague and blurry predictions, which is not a proper measurement of perceptual similarity. While SSIM was originally proposed to address the problem, Zhang et al. (2018) shows that their proposed LPIPS metric aligns better to human perception.

**Learning Strategy**   All models are trained with ADAM optimizer (Kingma & Ba, 2014) with $\mathcal{L}_1 + \mathcal{L}_2$ loss. Learning rate decay and scheduled sampling (Bengio et al., 2015) are used to ease training. Scheduled sampling is started once the model does not improve in 20 epochs (in term of validation loss), and the sampling ratio is decreased linearly from 1 until it reaches zero (by $2 \times 10^{-4}$ each epoch for Moving-MNIST-2 and $5 \times 10^{-4}$ for KTH). Learning rate decay is further activated if the loss does not drop in 20 epochs, and the rate is decreased exponentially by 0.98 every 5 epochs.

**Hyper-parameters Selection**   We perform a wide range of hyper-parameters search for Conv-TT-LSTM to identify the best model, and Table 1 summarizes our search values. The initial learning rate

| Kernel size | Initial learning rate | Tensor order | Tensor rank | Time steps |
|---|---|---|---|---|
| $\{3, 5\}$ | $\{1e\text{-}4, 5e\text{-}3, 1e\text{-}3\}$ | $\{1, 2, 3, 5\}$ | $\{4, 8, 16\}$ | $\{1, 3, 5\}$ |

Table 1: Hyper-parameters search values for Conv-TT-LSTM experiments.

of $10^{-3}$ is found for the models of kernel size 3 and $10^{-4}$ for the models of kernel size 5. We found that Conv-TT-LSTM models suffer from exploding gradients when learning rate is high (e.g. $10^{-3}$ in our experiments), therefore we also explore various gradient clipping values and select 1 for all Conv-TT-LSTM models. All hyper-parameters are selected using the best validation performance.

## 5.1 MOVING-MNIST-2 DATASET

The Moving-MNIST-2 dataset is generated by moving two digits of size $28 \times 28$ in MNIST dataset within a $64 \times 64$ black canvas. These digits are placed at a random initial location, and move with constant velocity in the canvas and bounce when they reach the boundary. Following Wang et al. (2018a), we generate 10,000 videos for training, 3,000 for validation, and 5,000 for test with default parameters in the generator[1]. All our models are trained to predict 10 frames given 10 input frames.

| Method | (10 -> 10) MSE | (10 -> 10) SSIM | (10 -> 10) LPIPS | (10 -> 30) MSE | (10 -> 30) SSIM | (10 -> 30) LPIPS | # params. |
|---|---|---|---|---|---|---|---|
| ConvLSTM (Xingjian et al., 2015) | 25.22 | 0.713 | - | 38.13 | 0.595 | - | 7.58M |
| CDNA (Finn et al., 2016) | 23.78 | 0.728 | - | 34.74 | 0.609 | - | - |
| VPN (Kalchbrenner et al., 2017) | 15.65 | 0.870 | - | 31.64 | 0.620 | - | - |
| E3D-LSTM (Wang et al., 2018b) | **10.08** | 0.910 | - | - | - | - | ≈15M |
| PredRNN++ (original) [2] | 11.35 | 0.898 | - | 22.24 | 0.814 | - | 15.05M |
| PredRNN++ (retrained) [3] | 10.29 | 0.913 | 59.51 | **20.53** | 0.834 | 139.9 | 15.05M |
| ConvLSTM (baseline) | 18.17 | 0.882 | 67.13 | 33.08 | 0.806 | 140.1 | 3.97M |
| Conv-TT-LSTM-FW (ours) | 14.29 | 0.906 | 48.29 | 28.88 | 0.831 | 104.1 | 2.65M |
| Conv-TT-LSTM-SW (ours) | 12.96 | **0.915** | **40.54** | 25.81 | **0.840** | **90.38** | 2.69M |

Table 2: Comparison of 10 and 30 frames prediction on Moving-MNIST-2 test set, where lower MSE values (in $10^{-3}$) / higher SSIM / lower LPIPS values (in $10^{-3}$) indicate better results. All our models use kernel size 5: Conv-TT-LSTM-FW has hyperparameters as (order 1, steps 3, ranks 8), and Conv-TT-LSTM-SW has hyperparameters as (order 3, steps 3, ranks 8).

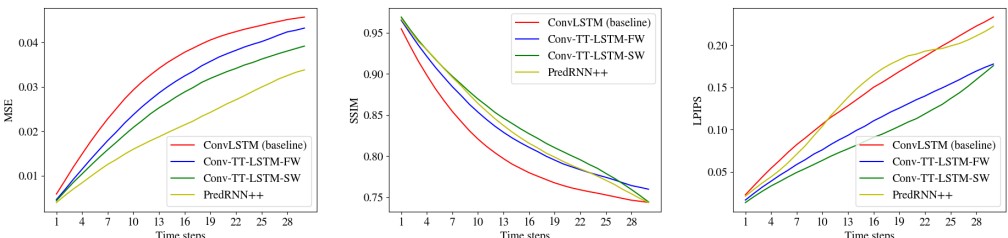

Figure 2: Frame-wise comparison in MSE, SSIM and PIPS on Moving-MNIST-2. For MSE and LPIPS, lower curves denote higher quality; while for SSIM, higher curves imply better quality.

**Multi-Steps Prediction** Table 2 reports the average statistics for 10 and 30 frames prediction, and Figure 2 shows comparison of per-frame statistics for PredRNN++ model, ConvLSTM baseline and our proposed Conv-TT-LSTM models. **(1)** Our Conv-TT-LSTM models consistently outperform the

---

[1] https://github.com/jthsieh/DDPAE-video-prediction/blob/master/data/moving_mnist.py
[2]The results are cited from the original paper, where the miscalculation of MSE is corrected in the table.
[3]The results are reproduced from https://github.com/Yunbo426/predrnn-pp with the same datasets in this paper. The original implementation crops each frame into patches as the input to the model. We find out such pre-processing is unnecessary and the performance is better than the original paper.

12-layer ConvLSTM baseline for both 10 and 30 frames prediction with fewer parameters; **(2)** The Conv-TT-LSTMs outperform previous approaches in terms of SSIM and LPIPS (especially on 30 frames prediction), *with less than one fifth of the model parameters.*

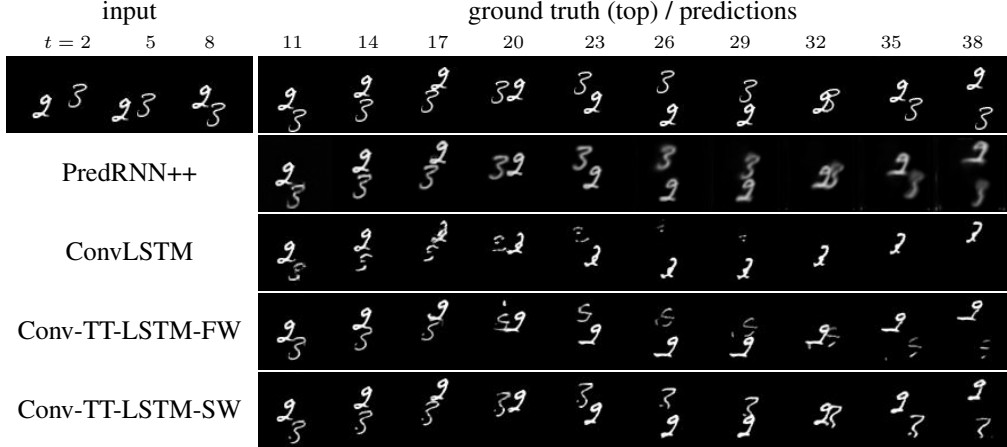

Figure 3: 30 frames prediction on Moving-MNIST given 10 input frames. Every 3 frames are shown.

We reproduce the PredRNN++ model (Wang et al., 2018a) from their source code[2], and we find that **(1)** The PredRNN++ model tends to output vague and blurry results in long-term prediction (especially after 20 steps). **(2)** and our Conv-TT-LSTMs are able to produce sharp and realistic digits over all steps. An example of comparison for different models is shown in Figure 3. The visualization is consistent with the results in Table 2 and Figure 2.

| Method | (10 -> 30) | | | # parameters |
| | MSE($\times 10^{-3}$) | SSIM | LPIPS | |
|---|---|---|---|---|
| Baseline ConvLSTM (4-layers model) | 37.19 | 0.791 | 184.2 | 11.48M |
| Conv-TT-LSTM-FW (4-layers model) | **31.46** | **0.819** | 112.5 | **5.65M** |
| Baseline ConvLSTM ($\mathcal{L}_1$ loss only) | 33.96 | 0.805 | 184.4 | 3.97M |
| Conv-TT-LSTM-FW ($\mathcal{L}_1$ loss only) | **30.27** | **0.827** | 118.2 | **2.65M** |
| Baseline ConvLSTM (teacher forcing) | 36.95 | 0.802 | 135.1 | 3.97M |
| Conv-TT-LSTM-FW (teacher forcing) | **34.84** | **0.807** | 128.4 | **2.65M** |
| Baseline ConvLSTM (our strategy) | 33.08 | 0.806 | 140.1 | 3.97M |
| Conv-TT-LSTM-FW (our strategy) | **28.88** | **0.831** | **104.1** | **2.65M** |

Table 3: Evaluation of ConvLSTM and our Conv-TT-LSTM under the ablated experimental settings.

**Ablation Study** To understand whether our proposed Conv-TT-LSTM universally improves upon ConvLSTM (i.e. not tied to specific architecture, loss function and learning schedule), we perform three ablation studies: **(1)** Reduce the number of layers from 12 layers to 4 layers (same as Xingjian et al. (2015) and Wang et al. (2018a)); **(2)** Change the loss function from $\mathcal{L}_1 + \mathcal{L}_2$ to $\mathcal{L}_1$ only; **(3)** Disable the scheduled sampling and use teacher forcing during training process. We evaluate the ConvLSTM baseline and our proposed Conv-TT-LSTM in these three settings, and summarize their comparisons in Table 3. The results show that our proposed Conv-TT-LSTM outperforms ConvLSTM consistently for all settings, i.e. the Conv-TT-LSTM model improves upon ConvLSTM in a board range of setups, which is not limited to the certain setting used in our paper. These ablation studies further show that our setup is optimal for predictive learning in Moving-MNIST-2.

## 5.2 KTH ACTION DATASET

KTH action dataset (Laptev et al., 2004) contains videos of 25 individuals performing 6 types of actions on a simple background. Our experimental setup follows Wang et al. (2018a), which uses

persons 1-16 for training and 17-25 for testing, and each frame is resized to $128 \times 128$ pixels. All our models are trained to predict 10 frames given 10 input frames. During training, we randomly select 20 contiguous frames from the training videos as a sample and group every 10,000 samples into one epoch to apply the learning strategy as explained at the beginning of this section.

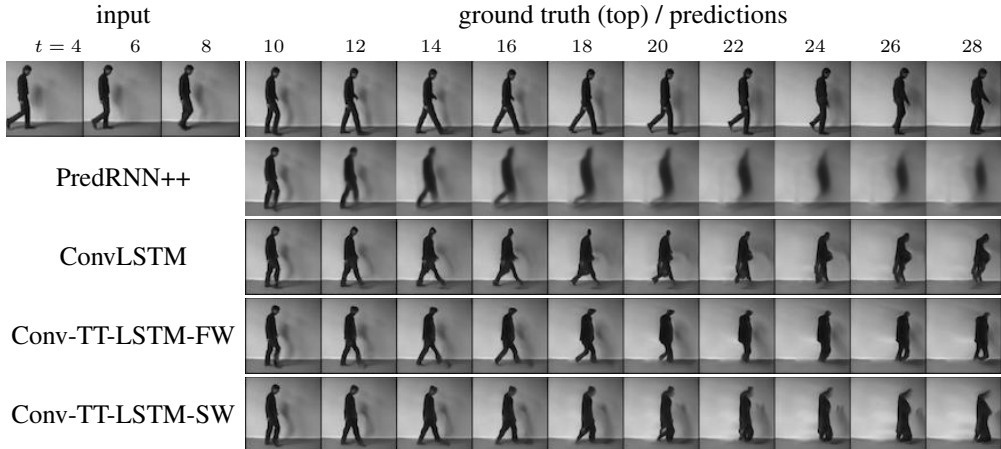

Figure 4: 20 frames prediction on KTH given 10 input frames. Every 2 frames are shown.

| Method | (10 -> 20) | | | (10 -> 40) | | | # Parameters |
|---|---|---|---|---|---|---|---|
| | PSNR | SSIM | LPIPS | PSNR | SSIM | LPIPS | |
| ConvLSTM (Xingjian et al., 2015) | 23.58 | 0.712 | - | 22.85 | 0.639 | - | 7.58M |
| MCNET (Villegas et al., 2017) | 25.95 | 0.804 | - | - | - | - | - |
| E3D-LSTM (Wang et al., 2018b) | **29.31** | 0.879 | - | **27.24** | 0.810 | - | ≈15M [4] |
| PredRNN++ (original)[2] | 28.46 | 0.865 | - | 25.21 | 0.741 | - | 15.05M |
| PredRNN++(retrained)[3] | 28.62 | 0.888 | 228.9 | 26.94 | 0.865 | 279.0 | |
| ConvLSTM-12 (baseline) | 28.21 | 0.903 | 137.1 | 26.01 | 0.876 | 201.3 | 3.97M |
| Conv-TT-LSTM-FW (ours) | 28.46 | **0.907** | 134.8 | 26.42 | **0.882** | 196.0 | 2.65M |
| Conv-TT-LSTM-SW (ours) | 28.36 | **0.907** | **133.4** | 26.11 | **0.882** | **191.2** | 2.69M |

Table 4: Evaluation of multi-steps prediction on KTH dataset, where higher PSNR or SSIM values indicate better predictive results. For Conv-TT-LSTM-FW, the reported model has hyperparameters (order 1, steps 3, ranks 8); and Conv-TT-LSTM-SW use hyperparameters (order 3, steps 3, ranks 8).

**Results** In Table 4, we report the evaluation on both 20 and 40 frames prediction. **(1)** Our models are consistently better than the ConvLSTM baseline for both 20 and 40 frames prediction. **(2)** While our proposed Conv-TT-LSTMs achieve lower SSIM value compared to the state-of-the-art models in 20 frames prediction, they outperform all previous models in LPIPS for both 20 and 40 frames prediction. An example of the predictions by the baseline and Conv-TT-LSTMs is shown in Figure 3.

## 6 CONCLUSION

In this paper, we proposed *convolutional tensor-train decomposition* to factorize a large convolutional kernel into a set of smaller *core tensors*. We applied this technique to efficiently construct *convolutional tensor-train LSTM* (Conv-TT-LSTM), a high-order spatio-temporal recurrent model whose parameters are represented in tensor-train format. We empirically demonstrated that our proposed Conv-TT-LSTM outperforms standard ConvLSTM and produce better/comparable results compared to other state-of-the-art models with fewer parameters. Utilizing the proposed model for high-resolution videos is still challenging due to gradient vanishing or explosion. Future direction will include investigating other training strategies or a model design to ease the training process.

---

[4] Wang et al. (2018b) mentions that the number of parameters is similar to PredRNN++ (Wang et al., 2018a).

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

# Appendix: Convolutional Tensor-Train LSTM for Long-Term Video Prediction

## A  PROOF OF THE SEQUENTIAL ALGORITHMS IN SECTION 3

In this section, we prove the sequential algorithms in Eq. (3) for tensor-train decomposition (1) and Eq. (6) for convolutional tensor-train decomposition (4) both by induction.

**Proof of Eq. (3)**  For simplicity, we denote the standard tensor-train decomposition in Eq. (1) as $\mathcal{T} = \mathsf{TTD}(\{\mathcal{T}^{(l)}\}_{l=1}^m)$, then Eq. (2) can be rewritten as Eq. (12) since $R_0 = 1$ and $v_1^{(0)} = 1$.

$$v = \sum_{r_0=1}^{R_0} \sum_{i_1=1}^{I_1} \cdots \sum_{i_m=1}^{I_m} \mathsf{TTD}\left(\{\mathcal{T}^{(l)}\}_{l=1}^m\right)_{i_1,\cdots,i_m} v_{r_0}^{(0)} \left(u^{(1)} \otimes \cdots \otimes u^{(m)}\right)_{i_1,\cdots,i_m} \tag{12}$$

$$= \sum_{r_0=1}^{R_0} \sum_{i_1=1}^{I_1} \cdots \sum_{i_m=1}^{I_m} \left(\sum_{r_1=1}^{R_1} \cdots \sum_{r_{m-1}=1}^{R_{m-1}} \mathcal{T}_{i_1,r_0,r_1}^{(1)} \cdots \mathcal{T}_{i_m,r_{m-1},r_m}^{(m)}\right) v_{r_0}^{(0)} u_{i_1}^{(1)} \cdots u_{i_m}^{(m)} \tag{13}$$

$$\begin{aligned} = & \sum_{r_1=1}^{R_1} \sum_{i_2=1}^{I_2} \cdots \sum_{i_m=1}^{I_m} \left(\sum_{r_2=1}^{R_2} \cdots \sum_{r_{m-1}=1}^{R_{m-1}} \mathcal{T}_{i_2,r_1,r_2}^{(2)} \cdots \mathcal{T}_{i_m,r_{m-1},r_m}^{(m)}\right) \\ & \qquad\qquad\qquad\qquad \left(\sum_{r_0=1}^{R_0} \sum_{i_1=1}^{I_1} \mathcal{T}_{i_1,r_0,r_1}^{(1)} v_{r_0}^{(0)} u_{i_1}^{(1)}\right) u_{i_2}^{(2)} \cdots u_{i_m}^{(m)} \end{aligned} \tag{14}$$

$$= \sum_{r_1=1}^{R_1} \sum_{i_2=1}^{I_2} \cdots \sum_{i_m=1}^{I_m} \mathsf{TTD}\left(\{\mathcal{T}^{(l)}\}_{l=2}^m\right)_{i_1,\cdots,i_m} v_{r_1}^{(1)} \left(u^{(2)} \otimes \cdots \otimes u^{(m)}\right)_{i_2,\cdots,i_m} \tag{15}$$

where $R_0 = 1$, $v_1^{(0)} = 1$ and the sequential algorithm in Eq. (3) is performed at Eq. (14).

**Proof of Eq. (6)**  For simplicity, we denote the convolutional tensor-train decomposition in Eq. (4) as $\mathcal{T} = \mathsf{CTTD}(\mathcal{T}^{(l)})_{l=1}^m$, then Eq. (5) can be rewritten as (16) since $\mathcal{V}^{(m)}$ is an all zeros tensor.

$$\mathcal{V}_{:,:,r_0} = \begin{aligned} & \sum_{l=1}^{m} \sum_{r_l=1}^{R_l} \mathsf{CTTD}\left(\{\mathcal{T}^{(t)}\}_{t=1}^l\right)_{:,:,r_l,r_0} * \mathcal{U}_{:,:,r_l}^{(l)} + \\ & \sum_{r_m=1}^{R_m} \mathsf{CTTD}\left(\{\mathcal{T}^{(t)}\}_{t=1}^m\right)_{:,:,r_m,r_0} * \mathcal{V}_{:,:,r_m}^{(m)} \end{aligned} \tag{16}$$

$$= \begin{aligned} & \sum_{l=1}^{m-1} \sum_{r_l=1}^{R_l} \mathsf{CTTD}\left(\{\mathcal{T}^{(t)}\}_{t=1}^l\right)_{:,:,r_l,r_0} * \mathcal{U}_{:,:,r_l}^{(l)} + \\ & \sum_{r_m=1}^{R_m} \mathsf{CTTD}\left(\{\mathcal{T}^{(t)}\}_{t=1}^m\right)_{:,:,r_m,r_0} * \left(\mathcal{U}_{:,:,r_m}^{(m)} + \mathcal{V}_{:,:,r_m}^{(m)}\right) \end{aligned} \tag{17}$$

Note that the second term in Eq. (17) can now be simplified as

$$\sum_{r_m=1}^{R_m} \mathsf{CTTD}\left(\{\mathcal{T}^{(t)}\}_{t=1}^m\right)_{:,:,r_m,r_0} * \left(\mathcal{U}_{:,:,r_m}^{(m)} + \mathcal{V}_{:,:,r_m}^{(m)}\right) \tag{18}$$

$$= \sum_{r_m=1}^{R_m} \left(\sum_{r_1=1}^{R_1} \cdots \sum_{r_{m-1}=1}^{R_{m-1}} \mathcal{T}_{:,:,r_1,r_0}^{(1)} * \cdots * \mathcal{T}_{:,:,r_m,r_{m-1}}^{(m)}\right) * \left(\mathcal{U}_{:,:,r_m}^{(m)} + \mathcal{V}_{:,:,r_m}^{(m)}\right) \tag{19}$$

$$= \sum_{r_{m-1}=1}^{R_{m-1}} \left(\sum_{r_1=1}^{R_1} \cdots \sum_{r_{m-1}=1}^{R_{m-1}} \mathcal{T}_{:,:,r_1,r_0}^{(1)} * \cdots * \mathcal{T}_{:,:,r_{m-1},r_{m-2}}^{(m-1)}\right) *$$
$$\left[\sum_{r_m=1}^{R_m} \mathcal{T}_{:,:,r_m,r_{m-1}}^{(m)} * \left(\mathcal{U}_{:,:,r_m}^{(m)} + \mathcal{V}_{:,:,r_m}^{(m)}\right)\right] \tag{20}$$

$$= \sum_{r_{m-1}=1}^{R_{m-1}} \mathsf{CTTD}\left(\{\mathcal{T}^{(t)}\}_{t=1}^{m-1}\right)_{:,:,r_{m-1},r_0} * \mathcal{V}_{:,:,r_{m-1}}^{(m-1)} \tag{21}$$

where the sequential algorithm in Eq. (5) is performed to achieve Eq. (21) from Eq. (20). Plugging Eq. (21) into Eq. (17), we reduce Eq. (17) back to the form as Eq. (16).

$$\mathcal{V}_{:,:,r_0} = \sum_{l=1}^{m-1} \sum_{r_l=1}^{R_l} \mathsf{CTTD}\left(\{\mathcal{T}^{(t)}\}_{t=1}^l\right)_{:,:,r_l,r_0} * \mathcal{U}_{:,:,r_l}^{(l)} +$$
$$\sum_{r_m=1}^{R_m} \mathsf{CTTD}\left(\{\mathcal{T}^{(t)}\}_{t=1}^{m-1}\right)_{:,:,r_{m-1},r_0} * \mathcal{V}_{:,:,r_{m-1}}^{(m-1)} \tag{22}$$

which completes the induction.

## B  SUPPLEMENTARY MATERIAL OF THE EXPERIMENTS

All experiments use a stack of 12-layers of ConvLSTM or Conv-TT-LSTM with 32 channels for the first and last 3 layers, and 48 channels for the 6 layers in the middle. A convolutional layer is applied on top of all LSTM layers to compute the predicted frames, followed by an optional sigmoid function (In the experiments, we add sigmoid for KTH dataset but not for Moving-MNIST-2). Additionally, two skip connections performing concatenation over channels are added between (3, 9) and (6, 12) layers as is shown in Figure 5.

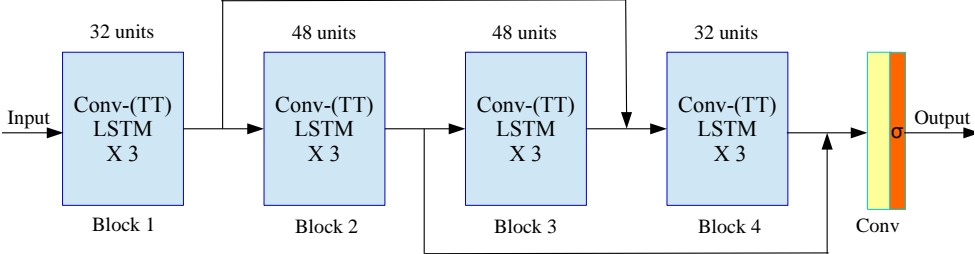

Figure 5: Illustration of the network architecture for the 12-layers model used in the experiments.

In this section, we provide additional results on the visual comparison between our proposed Conv-TT-LSTM and baseline ConvLSTM.

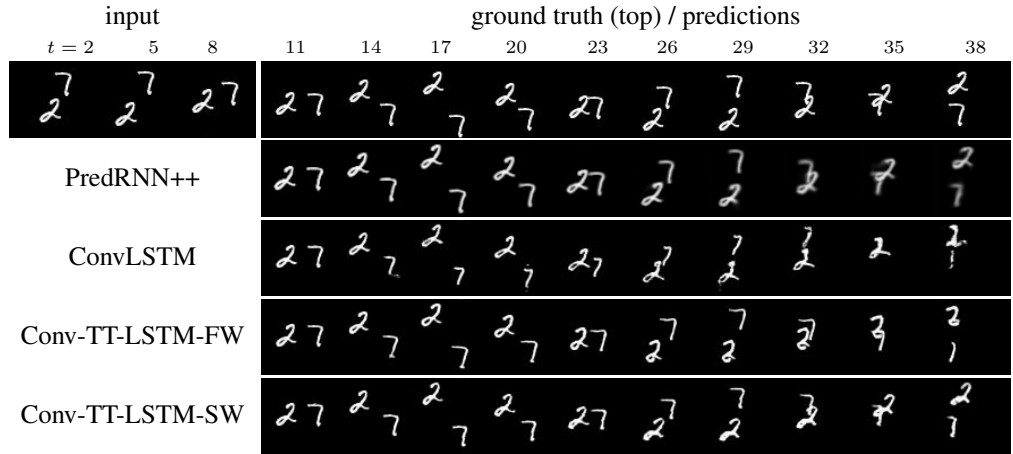

Figure 6: 30 frames prediction on Moving-MNIST given 10 input frames. Every 3 frames are shown.

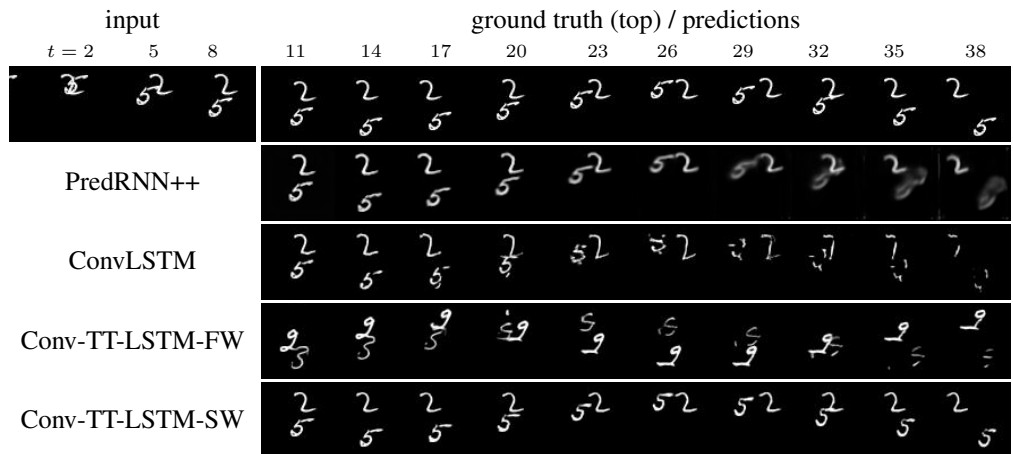

Figure 7: 30 frames prediction on Moving-MNIST given 10 input frames. Every 3 frames are shown.

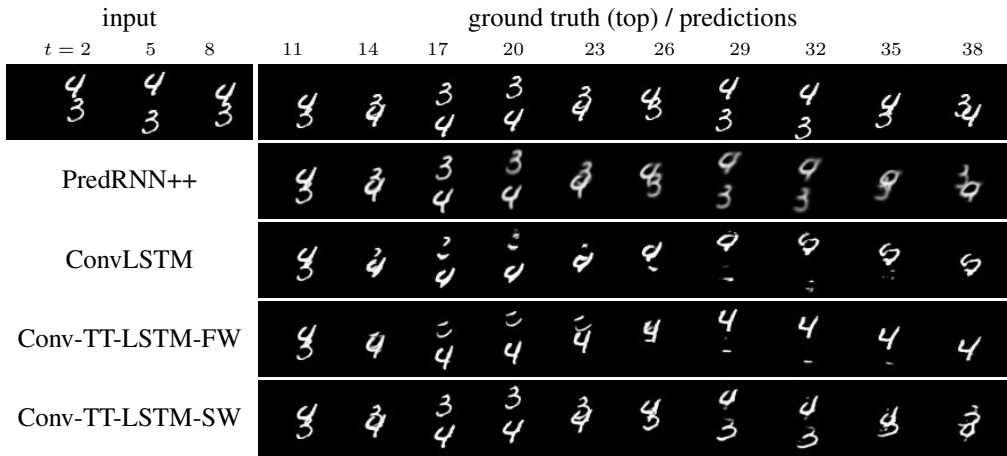

Figure 8: 30 frames prediction on Moving-MNIST given 10 input frames. Every 3 frames are shown.

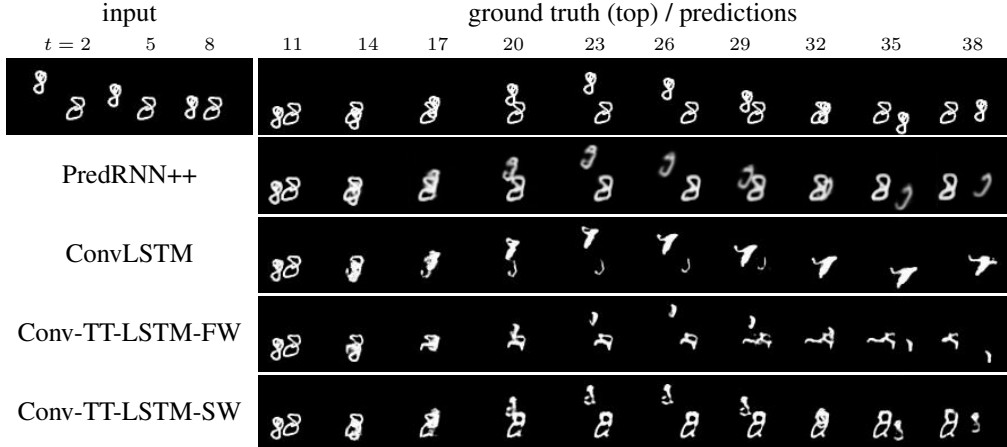

Figure 9: 30 frames prediction on Moving-MNIST given 10 input frames. Every 3 frames are shown.

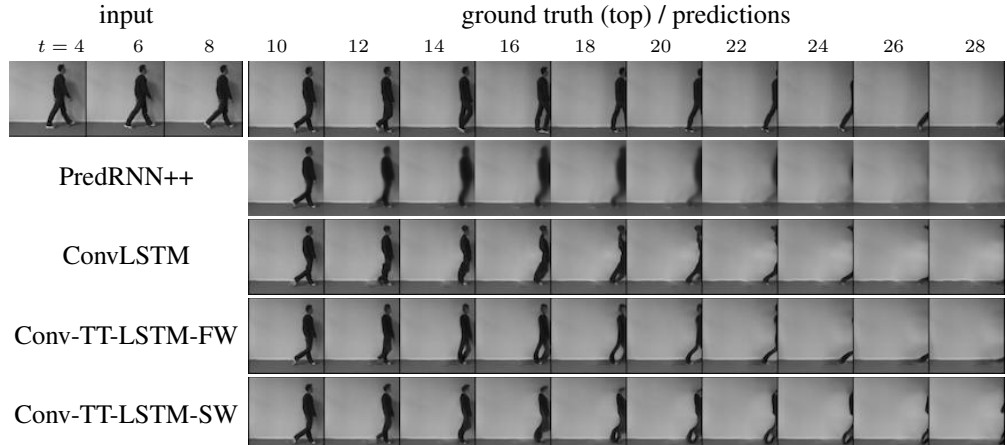

Figure 10: 20 frames prediction on KTH given 10 input frames. Every 2 frames are shown.

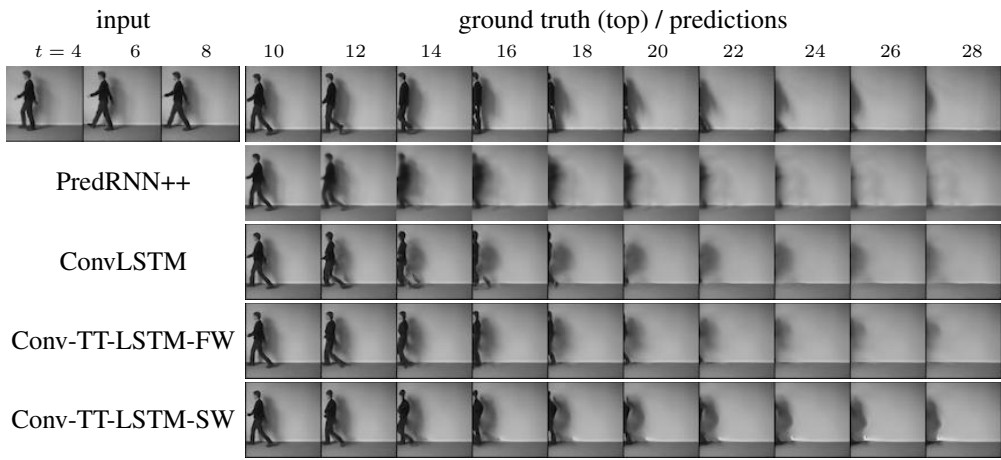

Figure 11: 20 frames prediction on KTH given 10 input frames. Every 2 frames are shown.

