# OpenReview forum: "Convolutional Tensor-Train LSTM for Long-Term Video Prediction"
_ICLR.cc/2020/Conference — Reject_

### Official Review · AnonReviewer1 · 2019-10-23
**Official Blind Review #1**

**Rating:** 3

**Review:**

Summary:
This paper proposes a method that saves memory and computation in the task of video prediction by low-rank tensor representations via tensor decomposition. The method is able to outperform standard convolutional lstm and other methods by using less parameters when testing it in the Moving MNIST and KTH datasets. The authors also present a proof to validate their method.


Pros:
+ Interesting method for decomposing tensors operations in convolutional architectures
+ Outperforms immediate baseline (Convolutional LSTM)

Weaknesses / comments:
- Weak experimental section
The authors mainly compare against Convolutional LSTM. The performance increase is there but the difference in parameters is not that significant in comparison to the performance. Needing fewer parameters is one of the claims in this paper and I am not fully convinced of the trade-off between the complexity of the model and the gain in parameter reduction / performance. In addition, the show videos do not look that much improved. The paper is also missing baselines from Villegas et al., 2017 and Denton et al., 2017 which both have available models for the KTH dataset.

- No videos provided
The paper does not provide any videos which is a must for video prediction papers. Judging the video quality from images in the paper is not easy, and also the used metrics have been shown to not be very objective in terms of video prediction quality or image generation in general.


Conclusion:
The proposed decomposition method is interesting, but the experimental section fails to convince me as to whether the methods performance validates the complicated formulations. My current score is between weak reject and reject so I will give a weak reject.

**Experience Assessment:**

I have published in this field for several years.

**Review Assessment: Checking Correctness Of Derivations And Theory:**

I assessed the sensibility of the derivations and theory.

**Review Assessment: Checking Correctness Of Experiments:**

I carefully checked the experiments.

**Review Assessment: Thoroughness In Paper Reading:**

I read the paper thoroughly.

---

> ### Author Response · Authors · 2019-11-15
> **Response to Official Blind Review #1**
>
> We thank the reviewer for the valuable comments.
>
> (1) Only comparing to Conv-LSTM, and the video quality not being improved.
> The improved results are provided in the revised version for both Moving-MNIST (in Table 2 and Figures 2, 3) and KTH (Table 4 and Figure 4). Our new results outperforms the state-of-the-art model, PredRNN++, on both SSIM and LPIPS metrics. In Figures 3, 5 and6-11, the visual examples show that our models produce much sharper predictions compared to PredRNN++.
>
> (2) Missing baselines from Villegas et al., 2017 and Denton et al., 2017.
> PSNR and SSIM scores of Villegas et al., 2017 has been included in Table 4, and are no better than our Conv-TT-LSTM models.
>
> (3) No video is provided.
> We included per-frame visualization (Fig. 3 and 5). We believe it is easier to judge the perceptual quality by looking at predicted results for each individual frame. We also added more samples in Fig. 6-11.

---

### Official Review · AnonReviewer2 · 2019-10-25
**Official Blind Review #2**

**Rating:** 3

**Review:**

This paper proposed a convolutional tensor-train (CTT) format based high-order and convolutional LSTM approach for long-term video prediction. This paper is well-motivated. Video data usually have high dimensional input, and the proposed method aims to explicitly take into account more than one hidden representation of previous frames - both lead to a huge number of parameters. Therefore, some sort of parameter reduction is needed. This paper considers two different types of operations - convolution and tensor-train (TT) decomposition - in an interleaved way. The basic model considered in this paper is a high-order variant of convolutional LSTM (convLSTM).

There exist several works using tensor decomposition methods including TT to compress a fully connected layer or a convolutional layer in neural nets, to break the memory bottleneck and accelerate computation. This paper takes a different direction - it further embeds convolution into the TT decomposition and thus defines a new type of tensor decomposition, termed convolutional tensor train (CTT) decomposition. CTT is used to represent the huge weight matrices arisen in the high-order convLSTM. To my best knowledge, this combination of convolution and TT decomposition is new.

The paper is well-written as the literature review is well done. Experimental results demonstrate improved performance over the convolutional LSTM baseline, a fewer number of parameters, and the qualitative results show sharp and clean digits. This improvement could be attributed to multiple causes: the high-order, the tensor decomposition-based compression, or the CTT. The authors also provide an ablation study, but it mainly concerns comparisons with ConvLSTM.

Despite the promising results, this paper is not ready for ICLR yet. Below is a list of suggested points needed to address:
(1) Yang et al 2017 claim that TT-RNN without convolution can also capture spatial and temporal dependence patterns in video modeling. This is an important baseline but missing in the current version of the paper.
(2) The justification of high-order modeling in long-term prediction. The first-order model also implicitly aggregates multiple past steps. It would be good to add more experimental evidence to support the necessity of the high-order.
(3) There exists some unjustified complexity for the CTT approach. How does it compare to TT for high-order ConvLSTM?

Perhaps, a more complete ablation study should include:
(1) LSTM with TT but without high-order and convolution
(2) LSTM with high-order and TT but without convolution
(3) ConvLSTM with TT
(4) ConvLSTM with CTT
(5) ConvLSTM with high-order and TT
(6) ConvLSTM with high-order and CTT

Question:
• How is the backpropagation done for the CTT core tensors?
• What is the error propagation issue of first-order methods and how does the high-order one not prone to it?

**Experience Assessment:**

I have read many papers in this area.

**Review Assessment: Checking Correctness Of Derivations And Theory:**

I assessed the sensibility of the derivations and theory.

**Review Assessment: Checking Correctness Of Experiments:**

I assessed the sensibility of the experiments.

**Review Assessment: Thoroughness In Paper Reading:**

I read the paper at least twice and used my best judgement in assessing the paper.

---

> ### Author Response · Authors · 2019-11-15
> **Response to Official Blind Review #2**
>
> Thank you for your efforts in reviewing our submission and your valuable suggestions of ablation studies.
>
> (A) Comparison to [Yang et al 2017] and necessity of Conv-LSTM (ablation studies (1) and (2)).
> The problem of video prediction is considerably more difficult than the one of video classification tackled in Yang et al, 2017: while only a single label is returned in video classification, video prediction requires producing all pixels for future frames. The work Yang et al, 2017 is based on fully-connected LSTM, which is generally not sufficient for video prediction. The original ConvLSTM paper [a] explains the necessity of convolutions for the video prediction problem. We believe this already answers your concern about using ConvLSTM instead of LSTM for video prediction.
>
> (B) Necessity of higher-order models (ablation study (4))
> We found that higher-order models generally perform better than first-order models, which justifies our preference for higher-order models. For example, if we reduce the reported higher-order model to first-order fixing other hyper-parameters unchanged, the PSNR will decrease by 1.1, and SSIM by 0.015.
>
> (C) Necessity of CTT (ablation studies (3) and (5)).
> We perform two ablation studies: the convolution filter size is fixed to 1 for CTT (which effectively reduce to TT) in Conv-TT-LSTM with higher-order and a single order. They corresponds to the ablation studies (3) and (5) as suggested. Unfortunately, both models are still under training but the validation curve ( https://postimg.cc/vx6rGnbH ) show that Conv-TT-LSTM without CTT is much worse than our proposed model. We expect them not much better than the ConvLSTM baseline. It suggests that convolutions in tensor-train is a very important component for capturing information in video prediction.
>
> (D) Question of backpropagation in CTT.
> In Equation (4), we derive an efficient sequential algorithm for using CTT in higher-order models. Therefore, in our current implementation, we use the built-in auto-differentiation for backpropagation, which effectively reverses the order of the forward iterations.
>
> (E) Question of error propagation issue.
> Compared to first-order models, higher-order models explicitly capture higher-order correlation, and therefore reduce the predictive error at each single step. As a result, the accumulation of the errors are slower over time, which benefits long-term prediction.
>
> [a] Xingjian, S. H. I., et al. "Convolutional LSTM network: A machine learning approach for precipitation nowcasting." Advances in neural information processing systems. 2015.

---

### Official Review · AnonReviewer4 · 2019-11-02
**Official Blind Review #4**

**Rating:** 3

**Review:**

This paper build a higher-order spatio-temporal model by means of combining Convolutional Tensor-Train Decomposition(CTTD) and ConvLSTM, and utilize the combination method to solve long-term video prediction problems. The CTTD factorizes a large convolutional kernel into a chain of smaller tensors, so as to relieve the difficult convergence and overfitting problems caused by too much model params.
Experiments on Moving-MNIST and KTH datasets show that the proposed method achieved better results than standard ConvLSTM, and in some way comparable with SOTA model. Ablation Studies are also provided.

Although it seems novel combing CTTD with ConvLSTM, the idea of CTTD and the combination mainly comes from [Yu et al.,2017] and [Yang et al.,2017]，this paper use the method in a new problem of video prediction,  I think the theoretical innovation is not enough for ICLR.
Although the experimental results were better than ConvLSTM(2015), but not as good as PredRNN++(2018), especially in terms of the MSE metrics. Since the prediction accuracy has not yet achieved, I don't think the reduction of model params is a matter of primary importance.  What’s more, Moving-MNIST and KTH are relatively simple datasets, video prediction on a more complicated datasets such as UCF101 will be more convincing.
Conclusion:
This paper is in some way novel, but not enough for ICLR, and the experiment results seems not enough convincing, so I will give a weak reject.


**Experience Assessment:**

I have read many papers in this area.

**Review Assessment: Checking Correctness Of Derivations And Theory:**

I carefully checked the derivations and theory.

**Review Assessment: Checking Correctness Of Experiments:**

I carefully checked the experiments.

**Review Assessment: Thoroughness In Paper Reading:**

I read the paper thoroughly.

---

> ### Author Response · Authors · 2019-11-15
> **Response to Official Blind Review #4**
>
> Thank you very much for your thoughtful comments.
>
> (1) Novelty of our work.
> (a) Yu et al. 2017 shows that higher-order models (compressed by standard tensor-train decomposition) perform better than first-order models in synthetic regression problems. However, their approach can not be easily extended to video prediction, since standard tensor-train cannot cope with the essential convolutional operations. We perform an ablation study regarding the necessity of convolutions in Tensor-Train (requested by Reviewer 2). The training is not finished yet, but the validation curve ( https://postimg.cc/vx6rGnbH ) shows that our higher-order model with Convolutional Tensor-Train is an important part of our proposed method.
>
> (2) Comparison against PredRNN++ (especially in terms of MSE).
> We added updated results in the revised version (Table 2 and 4). We found that our method produces sharper predictions, but MSE scores are lower (see Fig. 3, 4, 6-11). Therefore, we decided to add another metric, LPIPS, which better represents human perception. We discuss this issue in the experiment section. In the end, our methods outperform PredRNN++ on both SSIM and LPIPS metrics.
>
> (3) Necessity of model compression for higher-order models.
> For higher-order spatio-temporal models (such as higher-order Conv-LSTM considered in our paper), the parameters grow exponentially with the order, therefore the higher-order models cannot be built without model compression. In this paper, we show that even heavily compressed higher-order models can have better performance than (uncompressed) first-order models.
>
> [a] Zhang, Richard, et al. "The unreasonable effectiveness of deep features as a perceptual metric." Proceedings of the IEEE Conference on Computer Vision and Pattern Recognition. 2018.

---

### Author Response · Authors · 2019-11-14
**Updated paper with new results**

In this revised revision, we update the following parts of the paper:

1.  We add a perceptual metric, LPIPS [a] for all comparisons (Table 2-4). This metric is known to be close to human perception, compared to traditional metrics such as MSE and SSIM. We added a paragraph about a discussion of these metrics at the beginning of the experimental section.

2.  We reproduce the state-of-the-art ConvLSTM-based method, PredRNN++ [c], using their source code [b] on both Moving-MNIST and KTH datasets. By performing an additional hyper-parameter search, we obtained better performance than the numbers reported in the original paper. Compared to these results, our Conv-TT-LSTM outperforms PredRNN++ in both SSIM and LPIPS (reported in Table 2, 4, per-frame comparison in Fig 2). The visual samples show that our proposed methods are much sharper than PredRNN++ for long-term prediction on both datasets (Fig. 3, 4, 6-11).

3. Additional baseline, [Villegas et al., 2017] has been included in Fig. 4.

I believe this update answers the concerns about the quality of our method (reviewer 1 and 4). We will answer the rest of comments soon.

[a] Zhang, Richard, et al. "The unreasonable effectiveness of deep features as a perceptual metric." Proceedings of the IEEE Conference on Computer Vision and Pattern Recognition. 2018.
[b] https://github.com/Yunbo426/predrnn-pp
[c] Wang, Yunbo, et al. "Predrnn++: Towards a resolution of the deep-in-time dilemma in spatiotemporal predictive learning." arXiv preprint arXiv:1804.06300 (2018).

---

### Decision · Program_Chairs · 2019-12-19

**Decision:**

Reject

**Comment:**

This paper proposes Conv-TT-LSTM for long-term video prediction. The proposed method saves memory and computation by low-rank tensor representations via tensor decomposition and is evaluated in Moving MNIST and KTH datasets.

All reviews argue that the novelty of the paper does not meet the standard of ICLR. In the rebuttal, the authors polish the experiment design, which fails to change any reviewer’s decision.

Overall, the paper is not good enough for ICLR.